# Associations between Chronic Medical Conditions and Persistent Dietary Supplement Use: The US Military Dietary Supplement Use Study

**DOI:** 10.3390/nu16142253

**Published:** 2024-07-12

**Authors:** Joseph J. Knapik, Daniel W. Trone, Ryan A. Steelman, Harris R. Lieberman

**Affiliations:** 1Military Nutrition Division, US Army Research Institute of Environmental Medicine, 10 General Greene Ave., Building 42, Natick, MA 01760, USA; harris.r.lieberman.civ@health.mil; 2Deployment Health Research Department, Naval Health Research Center, Ryne Rd., Building 329, San Diego, CA 92152, USA; daniel.w.trone.civ@health.mil; 3Defense Centers for Public Health–Aberdeen, 8300 Ricketts Point Rd., Building E-2850, Aberdeen Proving Ground, MD 21010, USA

**Keywords:** anxiety, depression, gastroesophageal reflux disease, sleep apnea, hypercholesterolemia

## Abstract

This longitudinal study examined associations between chronic medical conditions (CMCs) and persistent dietary supplement (DS) use. On two separate occasions, 1.3 ± 0.2 years apart, military service members (SMs) (*n* = 5778) completed identical questionnaires concerning their DS use in the past 6 months and their demographic and lifestyle characteristics. Medical conditions were obtained from a medical surveillance system six months before the first questionnaire and during the period between questionnaires. Diagnoses were grouped into 19 major (largely systemic) and 9 specific CMCs. Conditions diagnosed in both periods (CMCs) were examined in relation to DS use reported on both questionnaires (persistent DS use). After adjustment for demographic and lifestyle factors, higher odds of persistent DS use were found in 7 of the 19 major CMCs and 5 of the 9 specific CMCs. SMs with a CMC had 1.25 (95% confidence interval [95%CI] = 1.10–1.41) higher adjusted odds of persistent DS use. The three specific CMCs with the highest adjusted odds of persistent DS use were anxiety (odds ratio [OR] = 2.30, 95%CI = 1.36–3.89), depression (OR = 2.12, 95%CI = 1.20–3.73), and gastroesophageal reflux disease (OR = 2.02, 95%CI = 1.02–4.04). Among DS categories, participants with a CMC had higher adjusted odds of persistent vitamins or mineral use (OR = 1.31, 95% CI = 1.12–1.53). Participants with CMCs had a higher prevalence of persistent DS use, especially individual vitamin and mineral use.

## 1. Introduction

A dietary supplement (DS) is a commercially available product consumed as an addition to the usual diet and includes vitamins, minerals, amino acids, herbs (botanicals), and a variety of other products [1]. Studies in military personnel [2,3] and civilians [4] have reported an increase in the prevalence of DS use, with the most recent data indicating that 56% of civilians [4] and 74% of service members (SMs) [3] used one or more DSs in the past year. The major reason both civilians and SMs report for using DSs is to enhance health [5,6,7,8]. Many DS users believe DSs are effective for the prevention or treatment of specific medical conditions such as cardiovascular diseases, hypercholesterolemia, osteoporosis, and depression [9,10,11,12]. However, there is limited evidence that DSs are efficacious in this regard, and favorable effects appear to be absent or modest [13,14,15,16]. Interestingly, 66–72% of American users state that they would continue to use DSs even if a government agency or the Food and Drug Administration reported the supplement was ineffective [17].

Several population-based studies have examined associations between DSs and medical conditions [18,19,20,21,22,23,24,25,26]. However, those cross-sectional investigations have depended on participant self-reports of medical problems, which could be subject to recall bias [27], and have reported on a limited number of medical conditions. We recently examined the association between DS use and a very comprehensive set of medical conditions in United States (US) military personnel [28]. We showed that those with clinically diagnosed medical conditions had higher odds of DS use compared to those without clinically diagnosed medical conditions [28]. In the current longitudinal study, we extend those findings by examining the association between clinically diagnosed chronic medical conditions and DS use over a longer period than previously reported in the literature. We hypothesized that participants with chronic medical conditions (CMCs) would have higher odds of persistent DS use.

## 2. Materials and Methods

Volunteers from a stratified random sample of US active-duty military SMs [3,29,30] completed identical surveys on two occasions. The surveys asked SMs about their DS use and their demographic and lifestyle characteristics. These data were combined with comprehensive medical records to examine CMCs in relation to persistent DS use (i.e., use reported on both occasions). The Naval Health Research Center Institutional Review Board approved this investigation, and SMs consented to participate by signing an informed consent document. Investigators adhered to policies and procedures for the protection of human subjects as prescribed by Department of Defense Instruction 3216.01, and the research was conducted in adherence to the provisions of 32 Code of Federal Regulations, Part 219.

### 2.1. Sampling Frame and Solicitation Procedures

There were two periods in this study: baseline (BL) and follow-up (FU). Details of the sampling frame, solicitation of SMs, subject recruitment flow chart, statistical power, and response bias in the BL and FU periods have been previously reported [3,29]. Briefly, investigators requested from the Defense Manpower Data Center a random sample of 200,000 SMs stratified by sex (88% male and 12% female) and branch of service (Army 36%, Air Force 24%, Marine Corps 15%, and Navy 25%) before the BL period. Recruitment of the randomly selected SMs into the BL period involved a maximum of 12 sequential contacts. These included an introductory postal letter (with a $1 bill designed to increase response rate [31,32]), a follow-up email message after 10 days, a postcard three weeks later, and up to seven emails and three postcard reminders evenly distributed across the time the survey was open. After this, contact with the SM ended. All postal and online contacts stated that at any time the SM could decline participation and be removed from the contact list. Recruitment in the BL period began in December 2018 and ended in August 2019.

As part of the BL informed consent, potential participants were informed there would be a FU that would involve the same procedures. Prior to the FU, the Defense Manpower Data Center identified SMs who were no longer on active duty, so they would not receive a FU request. SMs who volunteered for the BL and were still on active duty were asked to participate in the FU in a letter sent about 8 months after the BL, period closed. Solicitation procedures were the same as in the BL with 12 sequential contacts. Recruitment into the FU began in April 2020 and ended in December 2020.

### 2.2. Survey Description

The identical survey was used in the BL and FU periods and was based on previous ques-tionnaires of this type [33]. It was completed by participants online. The questionnaire included 96 generic DSs (e.g., multivitamins and minerals, individual vitamins and minerals, proteins, amino acids, herbals, joint health products, fish oils) and 91 brand-name products. The brand name products included some of those used in previous armed forces studies [2,6,34], but individual items were updated based on a review of DS inventories in the Army, Navy, and Air Force Exchange Services and General Nutrition Center stores on or near military installations before the start of the BL and FU periods. There were also open text fields on the questionnaire where SMs could include supplements not on the provided lists. For each listed DS, SMs were asked to estimate how frequently each supplement was used during the past 6 months (“never”, “once a month”, “once a week”, “2–6 times/week”, or “daily”). Demographic and lifestyle questions included items on date of birth (for age), gender, formal education, ethnicity, race, smoking and smokeless tobacco use, weekly aerobic and resistance training duration, and military service branch.

### 2.3. Medical Data

Once participants were identified by signing the informed consent form and completing the survey, the list of participants was sent to the Armed Forces Health Surveillance Division of the Defense Health Agency. From the Defense Medical Surveillance System database [35,36], the Defense Health Agency provided investigators with information on all medical encounters of volunteers. In the BL period, the medical conditions were provided for a 6-month period prior to the first question-naire. In the FU period, medical conditions were provided for the time between the two question-naires. Medical encounters in the Defense Medical Surveillance System were documented as In-ternational Classification of Diseases, Clinical Modification, and Revision 10 (ICD-10) codes, which were diagnosed and recorded by medical professionals. Encounters included those within military treat-ment facilities (i.e., Standard Ambulatory Data Record, Standard Inpatient Data Record, and Com-prehensive Ambulatory/Professional Encounter Record) as well as those outside these facilities (ci-vilian care) and paid for by the US Department of Defense (reimbursable) (i.e., Tricare Encounter Data-Institutional and Tricare Encounter Data-Noninstitutional).

### 2.4. Data Processing and Statistical Analysis

ICD-10 codes are a standard system used worldwide by medical professionals to classify medical conditions diagnosed during outpatient visits or hospitalization. The first letter of each code indicates a broad diagnostic category (e.g., infectious disease, circulatory diseases, injury and poisoning) and is followed by a series of numbers that provide more specific diagnoses within the broader category. For example, in the code G47.33, G denotes the major diagnostic category of diseases of the nervous system, the number 47 denotes sleep disorders, and 33 with this alphanumeric is the specific code for obstructive sleep apnea.

The ICD-10 code diagnoses of participants were grouped into the 28 conditions shown in Table 1. A CMC was defined as an ICD-10 code in one of the 28 code groups in both the BL and FU periods. A participant could have medical diagnoses in more than one CMC but were included only once within a single CMC. Major CMCs were 18 code groups representing the general disease and injury groups as defined in the ICD-10 codebook [37]. A separate code group included all ICD-10 codes (i.e., “any CMC”). Code groupings were developed for 201 specific CMCs, and those with at least 50 chronic cases (i.e., cases in both periods) were selected for analysis. Fifty cases provided a power of >80%. assuming a prevalence difference of about 40% in persistent DS use among those with and without a CMC. The 9 specific CMCs with ≥50 cases included hypercholesterolemia, sleep apnea, insomnia, depression, anxiety, hypertension, upper respiratory tract infection, gastroesophageal reflux disease (GERD), and osteoarthritis. Many of the specific code groupings (sleep apnea, insomnia, depression, anxiety, hypertension, GERD, and osteoarthritis) were those defined by the Armed Forces Health Surveillance Division of the Defense Health Service [38]. If these were not available for a specific condition, the code groupings were developed by the authors with assistance from medical professionals with specific areas of expertise.

Frequencies and proportions were calculated for each demographic and lifestyle characteristic at BL to describe the sample at the start of the investigation. Persistent DS use was defined as DS use reported in both the BL and FU periods. The prevalence (% ± standard error [SE]) of any persistent DS use was calculated by whether participants had a CMC in each of the 28 code groups. Univariate logistic regression models determined the odds of persistent DS use (a dependent variable) by having a CMC (independent variable). Multivariable logistic regression models adjusted the presence or absence of persistent DS use (dependent variable) by all demographic and lifestyle characteristics in addition to CMCs (independent variables). Demographic and lifestyle characteristics included in the multivariable analyses were gender, age, formal education, body mass index (BMI), ethnicity, race, smoking, smokeless tobacco use, weekly aerobic training duration, weekly resistance training duration, and military service branch. The relationship between chronic sleep apnea and BMI was further explored by examining the prevalence of sleep apnea at different levels of BMI using chi-square statistics to examine overall differences.

The prevalence of persistent use in various categories of DSs was also calculated by whether a participant had any CMC. Categories have been defined previously [28] and include multivitamins and minerals, individual vitamins and minerals, proteins and amino acids, purported prohormones, herbal substances, joint health products, and fish oils. Univariate and multivariable analyses were conducted for each of the 7 DS categories, as described above.

The prevalence of persistent DS use was determined based on the number of CMCs in the 18 major code groups (Table 1). For each participant, the number of CMCs within the major code groups was determined, and participants were placed into one of four groups: 0 (no CMCs), 1–2, 3–4, and ≥5 CMCs. Using odds ratios, Chi-squares, and linear trend statistics (Mantel-Haenszel test), the proportion of participants reporting persistent DSs use was compared between those without any CMC and those in each of the other 3 groups.

## 3. Results

From the initial sample frame of 200,000 SMs, 73% (n = 146,365) were successfully contacted (i.e., no returned postal mail) at BL, and of these, 26,680 (18.2%) signed the informed consent and completed the BL questionnaire. Of the 26,680 BL responders, 22,858 (85.7%) were still on active duty at the start of the FU and were successfully contacted at least once during the FU. Of these, 5778 completed the FU questionnaire, for a FU response rate of 25.3% (5778/22,858). The average ± SE follow-up time (time from BL to FU questionnaire completion) was 15.8 ± 2.0 months, with a range of 9.9–22.8 months. Compared to the original requested stratified sample, participants in the FU were more likely to be female (12% vs. 14%, *p* < 0.01) and consisted of more Air Force personnel with fewer personnel from other services (Air Force 39%, Army 31%, Marine Corps 10%, Navy 19%, *p* < 0.01).

Table 2 shows the demographic and lifestyle characteristics of the participants at BL. Participants were primarily white men, with an average ± standard deviation age of 35 ± 8 years. Ninety-three percent had some formal college education or a college degree. Participants varied substantially in time spent participating in weekly aerobic and resistance training exercise, and there was a relatively low proportion of smokers (15%) and smokeless tobacco users (10%).

The overall prevalence of persistent DS use was 67.6% (95% confidence interval = 66.4–68.8), and the overall prevalence of any CMC was 65.6% (95% confidence interval = 64.4–66.8). Table 3 shows associations between persistent DS use and CMCs. Overall, if any CMC was present, there were 1.25 (95% CI = 1.10–1.41) higher odds of persistent DS use after adjustment of demographic and lifestyle characteristics. In the adjusted analysis, there were significantly elevated odds of persistent DS use in 6 of the 18 major CMCs. In descending order, the major CMCs with the highest adjusted odds of persistent DS use were mental/behavioural disorders, digestive system diseases, endocrine/nutritional/metabolic disorders, nervous system diseases, musculoskeletal system diseases, and signs/symptoms/abnormal labs not otherwise specified. In the adjusted analysis, there were significantly elevated odds of persistent DS use in 5 of the 9 specific CMCs. In descending order, the specific CMCs with the highest odds of persistent DS use were anxiety, depression, GERD, hypercholesterolemia, and sleep apnea. The prevalence ± SE of chronic sleep apnea at BMI levels of <25.0, 25–29.9, and ≥30.0 kg/m^2^ was 2.0 ± 0.3, 5.5 ± 0.4, and 14.7 ± 1.2, respectively (*p* < 0.01). Among SM diagnosed with chronic depression, 28% were also diagnosed with chronic anxiety.

Table 4 shows the association between any CMC and persistent DS use by the DS categories. In the univariate analysis (unadjusted ORs), those with CMCs had higher odds of using multivitamin and minerals, individual vitamins and minerals, herbals, and joint health products, but lower use of proteins and amino acids. In the multivariable analysis, those with CMCs only had higher odds of individual vitamin or mineral use.

Table 5 shows the prevalence of persistent DS use by the number of CMCs. As the number of CMCs increased, so did the odds of persistent DS use.

## 4. Discussion

To our knowledge, this is the first study to examine associations between clinically diagnosed CMCs and persistent DS use. Overall, the odds of persistent DS use were 1.25 times higher among those with a CMC compared to those without one after adjustment for demographic and lifestyle factors. When individual CMCs were examined, the adjusted odds of persistent DS use were higher in virtually all major CMCs except congenital abnormalities, although this was statistically significant in only 6 of the 18 major categories. Of the nine specific CMCs examined, the adjusted odds of persistent DS use were higher for all nine and statistically significant for five of these. When various categories of DSs were examined, only individual vitamins and minerals had significantly higher adjusted odds of persistent DS use among those with a CMC. Also, the prevalence of persistent DS use rose in a linear manner as the number of major CMCs increased. In summary, these data indicate that among SMs, many CMCs are associated with persistent DS use, especially individual vitamin and mineral use, and a greater number of CMCs are associated with a higher prevalence of persistent DS use.

The higher adjusted odds of persistent vitamin or mineral use among those with a CMC may be associated with the wide variety of substances in this category and marketing claims and media attention that encourage use. Adequate intake of vitamins and minerals is needed to prevent nutritional deficiencies, and data from the National Health and Nutrition Examination Survey (NHANES) indicate that intake of some vitamins and minerals (notably vitamins A, C, D, and zinc) is below the estimated average requirement even when supplement use is considered [39]. Nonetheless, clinically diagnosed deficiencies are rare in military personnel [40,41], and excessive consumption of some vitamins and minerals can have detrimental effects [42].

Also of interest was the fact that persistent protein and amino acid users had lower odds of a CMC, although after adjustment for demographic and lifestyle factors, this association was attenuated and no longer statistically significant. We previously showed in the larger BL cohort (n = 26,680) that protein and amino acid users were more likely to be younger men who were more physically active, all factors that lower the risk of medical conditions [28].

The odds of persistent DS use were higher when specific CMCs were examined compared to when the major CMCs were examined. This was likely because the major categories contain many different types of diseases and injuries, some of which may have been associated with higher DS use and some not, while the specific CMCs involved more precise diagnoses. Nonetheless, many general-category CMCs were still associated with a high prevalence of persistent DS use. This suggests that other specific CMCs that the present study lacked sufficient statistical power to explore may have a high prevalence of persistent DS users. This should be explored in future studies with larger sample sizes.

### 4.1. Mental/Behavioral Disorders

Among those with and without CMCs, the largest difference in the adjusted odds of persistent DS use was within the major category of mental and behavioral disorders. Under this category were the specific disorders of chronic anxiety and depression, which had the highest adjusted odds of persistent DS use among the specific categories. It is estimated that 19% of American adults had an anxiety disorder in the past year, and 31% had the disorder at some point in their lives [43]. Depression is a leading cause of years lived with disability across the world [44], and in the US, 8% of adults had at least one depressive episode in 2021 [45]. Among all mental health disorders in the military during 2016–2020, anxiety and depressive disorders ranked second and third, respectively, after adjustment disorders [46]. Anxiety and depression are often comorbid [47], and in the current study, 28% of SMs had both chronic conditions, at least partly accounting for the similarity in the odds of persistent DS use.

Several cross-sectional studies have reported that individuals self-reporting anxiety symptoms have a higher prevalence of DS use [48,49,50,51,52]. Studies on self-reported depression are not as consistent. Some cross-sectional studies indicated higher DS use among those with self-reported depressive symptoms [48,52]. Satia-Abouta et al. [53] found that DSs use was higher among men self-reporting depression but not among women self-reporting depression. Friedman et al. [23] found little difference in overall DS use among those self-reporting depression vs. those not. The data in the current study indicated that SMs with clinically diagnosed chronic anxiety and depression were more likely to be persistent DS users.

Systematic reviews that included meta-analyses showed that B-vitamins and vitamin D have little influence on anxiety symptoms [54,55]. Most herbal substances that have been investigated for reducing generalized anxiety symptoms in randomized controlled trials showed little effect, but ginkgo biloba and ashwagandha have been reported to show some limited efficacy [56,57]. Similarly, systematic reviews of randomized controlled trials demonstrated that most vitamins and minerals, such as vitamin C, vitamin B-12, magnesium, and others, singly or in combination, have little or no effect on depressive symptoms. [54,58,59,60,61,62,63]. However, supplemental folate [64,65,66], vitamin E [55], zinc [67,68], and selenium [69] have been reported to reduce depressive symptoms on validated depressive surveys either by themselves or when combined with standard antidepressive medications. Most herbals that have been investigated have little influence on depressive symptoms [14,70,71], but randomized controlled trials on Saint John’s wort [72,73], saffron [74,75], and lavender [76,77] suggest limited efficacy.

### 4.2. Digestive Diseases

Among those with diagnosed digestive diseases, the adjusted odds of persistent DS use were the second highest among all the major CMCs, and under this category was GERD, which had the third highest adjusted odds of persistent DS use among the specific CMCs. The worldwide prevalence of physician diagnosed GERD is an estimated 14%, with a 21% prevalence in the US [78]. GERD is considerably less common in SMs, with one study of the entire military population reporting a clinically diagnosed prevalence of 1.0% [79], identical to that of the current study. The lower military prevalence is at least partly due to the younger age and predominate male sex of the military, since older age and female sex are GERD risk factors [78,79]. Satia-Abouta et al. found that individuals self-reporting “acid reflux disease” used a larger number of DSs [53], but no other investigations were found to have explored this condition in relation to DS use. Limited evidence in placebo-controlled trials supports the use of ginger to relieve GERD symptoms, although most other DSs tested show little efficacy in this regard [80].

### 4.3. Endocrine, Nutritional, and Metabolic Disease

Those diagnosed with chronic endocrine, nutritional, or metabolic disease had higher adjusted odds of DS use compared to those without a diagnosis in this category. Within this code group was chronic hypercholesterolemia, for which the adjusted odds of persistent DS use were also higher among those with this condition. Data from the NHANES indicated that rates of hypercholesterolemia have been declining since 2009, with the most recent data (2017–2020) indicating a prevalence of 36% in adults 20–44 years of age [81]. A study of the entire military population in 2007–2016 found a hypercholesterolemia prevalence of only 1.9% [82], similar to that of the current study. Previous cross-sectional investigations indicated that those with self-reported hypercholesterolemia had a higher prevalence of DS use [10,23] or used a greater number of DSs [53] compared to those without this condition.

Systematic reviews of randomized controlled trials have indicated that several DSs, including, for example, L-carnitine [83], coenzyme Q_10_ [84], flaxseed [85], fenugreek [86], and curcumin/turmeric [87], can very slightly lower blood or plasma total cholesterol levels, while other DSs such as fish oils [88], green coffee bean extract [89], and ginseng [90] have not been found effective in this regard. Large cholesterol lowering effects are found in red yeast rice [91,92], which can contain monacolin K, a compound identical to the active component in lovastatin, a cholesterol-lowering drug that reduces cholesterol synthesis [93]. There is very high variability in monacolin K content in commercial red yeast supplements [94]. Food and Drug Administration-approved statins have well-validated effects and can be administered in known dosages, making them a safer and more effective choice for cholesterol control [95].

### 4.4. Diseases of the Nervous System

SMs diagnosed with chronic nervous system disease had higher odds of DS use compared to those without a diagnosis in this category. This code grouping included sleep apnea, for which the adjusted odds of persistent DS use were also higher among those with this condition compared to those without. The incidence of sleep apnea in military personnel has been increasing over the years since at least 1997 [96]. This increase has been associated with the rise in SM’s body mass index [96], which has been increasing in a linear manner since at least 1975 [97,98], as well as factors related to deployment [96] and comorbid medical conditions such as traumatic brain injury and post-traumatic stress disorder that are often experienced by military personnel [96,99]. In the present study and other investigations [2,3,6], SMs with high BMI were more likely to be DS users, especially users of combination products often marketed for weight reduction [100]. Obesity may increase chronic sleep apnea by constricting airway passages or lung volume due to fat deposition or by altering neuromuscular control of upper airway passages [101].

### 4.5. Number of CMCs in Relation to Persistent DS Use

Cross-sectional data from the National Health Interview Study indicated that as the number of self-reported medical conditions increased, so did the proportion of DS users [18,20,102]. Cross-sectional data from a study of SMs reported similar results for clinically diagnosed medical conditions [28]. The present data expands on these findings by demonstrating that as the number of chronic, clinically diagnosed medical conditions increased, so did the prevalence of persistent DS use.

### 4.6. Strengths and Limitations

The medical database used in this study contained virtually complete information on diagnosed medical conditions experienced by SMs in the BL and FU periods. The study controlled for multiple demographic and lifestyle factors that could have confounded associations between CMCs and persistent DS use. Despite these strengths, there were limitations. Only 25.3% of the BL sample completed the FU survey, so there was a possibility of self-selection bias. Participants also differed from the requested stratified sample. As noted in detail in another paper [29], the FU cohort was more likely to be female and consisted of more Air Force personnel with fewer personnel from other services. Compared to those not participating in the larger sample frame, the FU cohort at BL was older (29 ± 7 vs. 35 ± 8), had more time in service (8 ± 6 vs. 12 ± 7), achieved higher formal educational levels (21% vs. 59% with college degrees), and were more likely to be officers (17% vs. 43%). Data regarding DS us, and demographic and lifestyle factors were self-reported and shared the usual limitations of these types of data, including recall bias, social desirability, errors in self-observation, and inadequate recall [103,104]. Persistent DS users were defined as reporting DS use on only two occasions, an average of about 16 months apart; a longer study (e.g., over several years) with more questionnaire administrations may have improved the identification of longer-term DS users. Similarly, CMCs were those diagnosed only twice, and diagnoses over a longer timeframe may more accurately identify chronic conditions. We had only enough statistical power to examine only nine specific CMCs, and a study with a larger sample size might be able to investigate a wider range of specific medical conditions. There were a large number of statistical tests performed, on the medical conditions examined here. The more statistical tests performed the higher the likelihood of a false positive or Type 1 error where the null hypothesis is rejected when it is correct.

## 5. Conclusions

This study demonstrated that SMs with CMCs were more likely to persistently use DSs over a >1 year period, especially individual vitamins, and minerals. Also, the prevalence of persistent DS use increased as the number of CMCs increased. These data support the hypothesis that SMs may be using DS to self-treat medical conditions, although the possibility that the DSs may be causing the CMC may also be considered. While a limited number of DSs exhibit efficacy regarding the prevention and/or treatment of some medical problems, meta-analyses of randomized controlled trials show that favorable effects are generally modest and limited. This study contributes to our understanding of the association between DS use and medical conditions by examining chronic medical conditions diagnosed by medical care providers and incorporating the full range of clinically diagnosed CMCs in relation to persistent DS use.

## Figures and Tables

**Table 1 nutrients-16-02253-t001:** ICD-10 Codes for Chronic Medical Conditions ^a^.

ICD-10 Code Grouping	CMCs	ICD-10 Codes	*n*	Prevalence of CMCs (% ± SE)
Any CMC	Any CMC	A00 through Y99	3792	65.6 ± 0.8
Major Code Groups	Infectious/Parasitic Diseases	A00 through B99	116	2.0 ± 1.3
	Neoplasms	C00 through D49	133	2.3 ± 1.3
	Diseases of the Blood and Blood Organs	D50 through D89	23	0.4 ± 1.3
	Endocrine, Nutritional and Metabolic Diseases	E00 trough E89	335	5.8 ± 1.3
	Mental and Behavioral Diseases	F01 trough F99	424	7.3 ± 1.3
	Diseases of the Nervous System	G00 through G99	639	11.1 ± 1.2
	Diseases of the Eye and Adnexa	H00 though H59	559	9.7 ± 1.3
	Diseases of the Ear and Mastoid Process	H60 through H95	118	2.0 ± 1.3
	Diseases of the Circulatory System	I00 through I99	200	3.5 ± 1.3
	Diseases of the Respiratory System	J00 through J99	518	9.0 ± 1.3
	Diseases of the Digestive System	K00 through K95	171	3.0 ± 1.3
	Diseases of the Skin and Subcutaneous Areas	L00 through L99	305	5.3 ± 1.3
	Diseases of the Musculoskeletal System	M00 through M99	1614	27.9 ± 1.1
	Diseases of the Genitourinary System	N00 through N99	154	2.7 ± 1.3
	Congenital Abnormalities	Q00 through Q99	20	0.3 ± 1.2
	Signs, Symptoms, and Abnormal Labs	R00 through R99	1411	24.4 ± 1.1
	Injury/Poisoning	S00 through T88	229	4.0 ± 1.3
	External Causes of Morbidity	V00 through Y99	66	1.1 ± 1.3
Specific Code Groups	Hypercholesterolemia	E78.00, E78.01, E78.1, E78.2, E78.3, E78.4, E78.5, E78.6	105	1.8 ± 1.3
	Sleep Apnea	G47.30, G47.31, G47.32, G47.33, G47.34, G47.35, G47.36, G47.37, G47.39	344	5.9 ± 1.3
	Insomnia	F51.01, F51.02, F51.03, F51.04, F51.05, F51.09, G47.00, G47.01, G47.09	59	1.0 ± 1.3
	Depression	F32.0, F32.1, F32.2, F32.3, F32.4, F32.5; F34.8, F32.81, F32.89, F32.9, F33.0, F33.1, F33.2, F33.3, F33.4, F33.40, F33.41, F33.42, F33.8, F33.9, F34.1	86	1.5 ± 1.3
	Anxiety	F40.00, F40.01, F40.02, F40.10, F40.11, F40.210, F40.218, F40.220, F40.228, F40.230, F40.231, F40.232, F40.233, F40.240, F40.241, F40.242, F40.243, F40.248, F40.290, F40.291, F40.298, F40.8, F40.9, F41.0, F41.1, F41.3, F41.8, F41.9, 42.2, F42.3, F42.4, F42.8,F42.9, F48.8	109	1.9 ± 1.3
	Hypertension	I10, I11.0, I11.9, I12.0, 12.9, I13.0, I13.10, I13.11, I13.2, I15.0, I15.1, I15.2, I15.8, I15.9, I16.0, I16.1, I16.9, I67.4, I87.301, I87.302, I87.303, I87.309, I87.311, I87.312, I87.319, I87.321, I87.322, I87.323, I87.329, I87.331, I87.332, I87.333, I87.339, G93.2	156	2.7 ± 1.3
	Upper Respiratory Tract Infection	J00 through J06.9, J30 through J39.9	217	3.8 ± 1.3
	Gastro-Esophageal Reflux Disease	K21.0, K21.9	58	1.0 ± 1.3
	Osteoarthritis	M15.0, M15.1, M15.2, M15.3, M15.4, M15.8, M15.9, M16.0, M16.10, M16.11, M16.12, M16.2, M16.30, M16.31, M16.32, M16.4, M16.50, M16.51, M16.52, M16.6, M16.7, M16.9, M17.0, M17.10, M17.11, M17.12, M17.2, M17.30, M17.31, M17.32, M17.4, M17.5, M17.9, M18.0, M18.10, M18.11, M18.12, M18.2, M18.30, M18.31, M18.32, M18.4, M18.50, M18.51, M18.52, M18.9, M19.011, M19.012, M19.019, M19.021, M19.022, M19.029, M19.031, M19.032, M19.039, M19.041, M19.042, M19.049, M19.071, M19.072, M19.079, M19.111, M19.112, M19.119, M19.121, M19.122, M19.129, M19.131, M19.132, M19.139, M19.141, M19.142, M19.149, M19.171, M19.172, M19.179, M19.211, M19.212, M19.219, M19.221, M19.222, M19.229, M19.231, M19.232, M19.239, M19.241, M19.242, M19.249, M19.271, M19.272, M19.279, M19.90, M19.91, M19.92, M19.93	57	1.0 ± 1.3

Abbreviations: CMC = chronic medical condition; ICD-10 = International Classification of Diseases, Clinical Modification, Revision 10. ^a^ A chronic medical condition was one that was diagnosed using these codes in both the baseline and follow-up periods.

**Table 2 nutrients-16-02253-t002:** Demographic and Lifestyle Characteristics of Active Duty Military Personnel at Baseline.

Variable	Strata	Sample Size (*n*)	Proportion of Sample (%)
Gender	Men	4972	86.1
	Women	806	13.9
Age	18–24 yr	581	10.1
	25–29 yr	948	16.4
	30–39 yr	2620	45.7
	≥40 yr	1578	27.6
Formal Education	Some High School/High School Graduate	438	7.6
	Some College	1962	34.0
	Bachelor/Graduate Degree	3378	58.5
Body Mass Index	<25.0 kg/m^2^	1758	31.0
	25.0–29.9 kg/m^2^	3094	54.5
	≥30.0 kg/m^2^	827	14.6
Ethnicity	Not Hispanic	5057	87.5
	Hispanic	721	12.5
Race	White	4515	78.1
	Black	491	8.5
	American Indian	37	0.6
	Asian	245	4.2
	Pacific Islander	28	0.5
	Other	159	2.8
	Multiple	303	5.2
Smoking	Never Smoked	3851	67.5
	Smoked but Quit	1023	17.9
	Smoker	833	14.6
Smokeless Tobacco	Never Used	4640	82.5
	Used but quit	452	8.0
	User	532	9.5
Alcohol Consumption	0 mL/day	1498	25.9
	0.23–24.85 mL/day	1404	24.3
	24.86–71.69 mL/day	1400	24.2
	>71.69 mL/day	1475	25.5
Aerobic Exercise	≤90 min/wk	1560	27.0
	91–180 min/wk	1685	29.2
	181–300 min/wk	1325	22.9
	>300 min/wk	1208	20.9
Resistance Exercise	<45 min/wk	1775	30.7
	46–135 min/wk	1466	25.4
	136–300 min/wk	1415	24.5
	>300 min/wk	1122	19.4
Service Branch	Air Force	2295	39.7
	Army	1807	31.3
	Marine Corps	574	9.9
	Navy	1102	19.1

**Table 3 nutrients-16-02253-t003:** Associations between Persistent DS Use and Clinically Diagnosed Chronic Medical Conditions.

Category	CMCs	CMC Present	*n*	Persistent DS Use Prevalence (% ± SE)	Unadjusted		Adjusted ^a^	
					Odds Ratio (95% CI)	*p*-Value	Odds Ratio (95% CI) ^a^	*p*-Value
Any CMC	Any CMC	Yes	3792	69.8 ± 0.7	1.33	<0.01	1.25	<0.01
		No	1986	63.5 ± 1.1	(1.19–1.49)		(1.10–1.41)	
Major CMCs	Infectious/Parasitic Diseases	Yes	116	69.8 ± 4.3	1.11	0.61	1.07	0.78
		No	5662	67.6 ± 0.6	(0.74–1.71)		(0.69–1.65)	
	Neoplasms	Yes	133	72.9 ± 3.9	1.30	0.19	1.23	0.32
		No	5645	67.5 ± 0.6	(0.89–1.93)		(0.82–1.86)	
	Diseases of Blood and Blood Organs	Yes	23	65.2 ± 9.9	0.90	0.80	1.04	0.94
		No	5755	67.6 ± 0.6	(0.38–2.24)		(0.38–2.81)	
	Endocrine, Nutritional and Metabolic Diseases	Yes	335	76.7 ± 2.3	1.62	<0.01	1.51	<0.01
		No	5443	67.1 ± 0.6	(1.25–2.11)		(1.14–2.00)	
	Mental and Behavioral Disorders	Yes	424	76.4 ± 2.1	1.60	<0.01	1.65	<0.01
		No	5354	66.9 ± 0.6	(1.27–2.02)		(1.28–2.11)	
	Diseases of the Nervous System	Yes	639	75.3 ± 1.7	1.52	<0.01	1.51	<0.01
		No	5139	66.7 ± 0.7	(1.26–1.84)		(1.22–1.86)	
	Diseases of the Eyes and Adnexa	Yes	559	70.8 ± 1.9	1.18	0.09	1.14	0.21
		No	5219	67.3 ± 0.6	(0.97–1.43)		(0.93–1.40)	
	Diseases of the Ear and Mastoid Process	Yes	118	73.7 ± 4.1	1.35	0.15	1.24	0.35
		No	5660	67.5 ± 0.6	(0.90–2.07)		(0.80–1.92)	
	Diseases of the Circulatory System	Yes	200	72.5 ± 3.2	1.27	0.14	1.30	0.15
		No	5578	67.5 ± 0.6	(0.92–1.75)		(0.91–1.84)	
	Diseases of the Respiratory System	Yes	518	71.2 ± 2.0	1.20	0.07	1.17	0.16
		No	5260	67.3 ± 0.6	(0.99–1.47)		(0.94–1.45)	
	Diseases of the Digestive System	Yes	171	75.4 ± 3.3	1.49	0.03	1.52	0.03
		No	5607	67.4 ± 0.6	(1.05–2.13)		(1.04–2.20)	
	Diseases of the Skin and Subcutaneous Tissue	Yes	305	69.8 ± 2.6	1.11	0.40	1.06	0.69
		No	5473	67.5 ± 0.6	(0.87–1.44)		(0.81–1.38)	
	Diseases of the Musculoskeletal System	Yes	1614	75.0 ± 1.1	1.64	<0.01	1.46	<0.01
		No	4164	64.8 ± 0.7	(1.44–1.86)		(1.27–1.68)	
	Diseases of the Genitourinary System	Yes	154	76.6 ± 3.4	1.59	0.02	1.44	0.09
		No	5624	67.4 ± 0.6	(1.10–2.34)		(0.95–2.20)	
	Congenital Abnormalities	Yes	20	65.0 ± 10.7	0.89	0.80	0.74	0.55
		No	5758	67.6 ± 0.6	(0.36–2.38)		(0.28–1.98)	
	Signs, Symptoms, and Abnormal Labs, NOS	Yes	1411	73.4 ± 1.2	1.43	<0.01	1.39	<0.01
		No	4367	65.8 ± 0.7	(1.25–1.64)		(1.21–1.61)	
	Injury and Poisoning	Yes	229	73.8 ± 2.9	1.36	0.04	1.25	0.18
		No	5549	67.4 ± 0.6	(1.01–1.85)		(0.90–1.73)	
	External Causes of Morbidity	Yes	66	69.7 ± 5.7	1.10	0.72	1.02	0.96
		No	5712	67.6 ± 0.6	(0.66–1.90)		(0.58–1.77)	
Specific CMCs	Hypercholesterolemia	Yes	105	76.2 ± 4.2	1.54	0.06	1.94	<0.01
		No	5673	67.5 ± 0.6	(0.98–2.43)		(1.18–3.20)	
	Sleep Apnea	Yes	344	76.2 ± 2.3	1.56	<0.01	1.67	<0.01
		No	5434	67.1 ± 0.6	(1.22–2.05)		(1.25–2.23)	
	Insomnia	Yes	59	72.9 ± 5.8	1.29	0.39	1.31	0.39
		No	5719	67.6 ± 0.6	(0.72–2.30)		(0.71–2.44)	
	Depression	Yes	86	80.2 ± 4.3	1.96	0.01	2.12	0.01
		No	5692	67.4 ± 0.6	(1.15–3.34)		(1.20–3.73)	
	Anxiety	Yes	109	82.6 ± 3.6	2.30	<0.01	2.30	<0.01
		No	5669	67.3 ± 0.6	(1.40–3.78)		(1.36–3.89)	
	Hypertension	Yes	156	73.1 ± 3.6	1.31	0.14	1.42	0.09
		No	5622	67.5 ± 0.6	(0.91–1.87)		(0.95–2.11)	
	Upper Respiratory Tract Infection	Yes	217	69.6 ± 3.1	1.10	0.53	1.14	0.43
		No	5561	67.6 ± 0.6	(0.82–1.41)		(0.83–1.56)	
	Gastroesophageal Reflux Disease	Yes	58	81.0 ± 5.2	2.06	0.03	2.02	0.05
		No	5720	67.5 ± 0.6	(1.07–3.98)		(1.02–4.01)	
	Osteoarthritis	Yes	57	78.9 ± 5.4	1.80	0.07	1.49	0.26
		No	5721	67.5 ± 0.6	(0.95–3.42)		(0.75–2.95)	

Abbreviations: 95%CI = 95% confidence interval; CMC = chronic medical condition; DS = dietary supplement; NOS = not otherwise specified; SE = standard error. ^a^ Adjusted for baseline gender, age, formal education, body mass index, race, ethnicity, weekly aerobic exercise duration, weekly resistance training duration, cigarette smoking, smokeless tobacco use, alcohol consumption, and service branch.

**Table 4 nutrients-16-02253-t004:** Associations between Any Chronic Medical Condition and Persistent DS Use by DS Category.

DS Category	Amy CMC Yes *n* = 3792 No *n* = 1986	Prevalence of Persistent DS Use % (95% CI)	Unadjusted		Adjusted ^a^	
			Odds Ratio (95% CI)	*p*-Value	Odds Ratio (95%CI)	*p*-Value
Multivitamin /Multimineral	Yes	37.2 (35.7–38.7)	1.21	<0.01	1.09	0.16
	No	32.8 (30.7–34.9)	(1.08–1.36)		(0.97–1.24)	
Individual Vitamins and Minerals	Yes	22.9 (21.6–24.2)	1.55	<0.01	1.31	<0.01
	No	16.1 (14.5–17.7)	(1.34–1.78)		(1.13–1.52)	
Proteins/Amino Acids	Yes	28.8 (27.4–30.2)	0.82	<0.01	0.88	0.06
	No	33.0 (30.9–35.1)	(0.73–0.92)		(0.77–1.01)	
Combination Products	Yes	27.4 (26.0–28.8)	1.03	0.65	1.11	0.16
	No	26.8 (24.9–28.7)	(0.91–1.16)		(0.96–1.28)	
Prohormones	Yes	2.0 (1.6–2.4)	1.02	0.92	0.93	0.72
	No	1.9 (1.3–2.5)	(0.69–1.51)		(0.61–1.42)	
Herbals	Yes	12.6 (11.5–13.7)	1.35	<0.01	1.14	0.17
	No	9.7 (8.4–11.0)	(1.13–1.61)		(0.95–1.38)	
Joint Health Products	Yes	7.4 (6.6–8.2)	1.32	0.02	1.17	0.19
	No	5.7 (4.7–6.7)	(1.05–1.65)		(0.93–1.49)	
Fish Oils	Yes	15.4 (14.3–16.5)	1.00	0.98	0.96	0.64
	No	15.5 (13.9–17.1)	(0.86–1.16)		(0.82–1.13)	

Abbreviations: 95%CI = 95% confidence interval; CMC = chronic medical condition; DS = dietary supplement; ^a^ Adjusted for baseline gender, age, formal education, body mass index, race, ethnicity, weekly aerobic exercise duration, weekly resistance training duration, cigarette smoking, smokeless tobacco use, alcohol consumption, and service branch.

**Table 5 nutrients-16-02253-t005:** Prevalence of Persistent DS Use by the Number of CMCs.

Any CMC (*n*)	Sample Size (*n*)	Prevalence of Persistent DS Users (% ± SE)	Odds Ratio (95% CI)	Chi Square *p*-Value	Linear Trend *p*-Value ^a^
None	1986	63.2 ± 1.1	1.00	Reference	<0.01
1–2	2562	69.6 ± 0.9	1.34 (1.18–1.52)	<0.01	
3–4	920	73.4 ± 1.5	1.61 (1.35–1.91)	<0.01	
≥5	310	79.0 ± 2.3	2.20 (1.65–2.93)	<0.01	

Abbreviations: 95%CI = 95% confidence interval; CMCs = chronic medical conditions; DSs = dietary supplements; SE = standard error; ^a^ Mantel-Haenszel statistic.

## Data Availability

The datasets presented in this article are not publicly available because of US government restrictions but can be obtained from the author on reasonable request and development of a Data Sharing Agreement.

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
