# Peer review of "Associations between Chronic Medical Conditions and Persistent Dietary Supplement Use: The US Military Dietary Supplement Use Study"

_nutrients, 2024, doi:10.3390/nu16142253_

Round 1

Reviewer 1 Report

Comments and Suggestions for Authors

This study was conducted in 5778 active duty US military SMs, and the longitudinal study examined the association between chronic diseases (CMC) and continuous dietary support (DS) use. It was found that SMs with CMC had higher effectiveness for continuous DS vitamin/mineral use. However, the study still has some shortcomings:

1. The significance of this study needs to be clarified. What is the purpose of exploring the relationship between dietary supplements and chronic diseases? What is its practical significance?

2. If persistent Down syndrome is defined as reporting only two uses of Down syndrome, with an average of about 16 months, it should be compared with the definitions in other articles.

3. The analysis is relatively simple, and the analysis of Down syndrome intake by age and sex for key chronic diseases can be analyzed.

4. References should be as new as possible.

Author Response

(Reviewer’s comments preceded by an “R”; author’s replies preceded by an “A”)

R: This study was conducted in 5778 active duty US military SMs, and the longitudinal study examined the association between chronic diseases (CMC) and continuous dietary support (DS) use. It was found that SMs with CMC had higher effectiveness for continuous DS vitamin/mineral use. However, the study still has some shortcomings:

A: Thank you for your summary.  We hope we have adequately addressed your comments and concerns below.

R1. The significance of this study needs to be clarified. What is the purpose of exploring the relationship between dietary supplements and chronic diseases? What is its practical significance?

A: As we state in the Introduction (Page 3-4, Lines 31-49), the primary reason both military personnel and civilians report for using dietary supplements (DSs) is to improve health.  The literature also indicates that many individuals believe that specific types of dietary supplements can be used to treat specific medical conditions, although when many of these DSs are tested for efficacy in double-blinded placebo controlled studies effects on specific medical conditions are absent or relatively modest.  On a practical level our goal to was extend finding from our previous on the relationship between DS use and medical conditions.  In that previous study we cross-sectionally examined associations between DS use and showed that those with many general and specific clinically diagnosed medical conditions had a higher odds of DS use compared to those not using DSs.  In the current study we wanted to see if those with chronic medical conditions (diagnosed on at least 2 occasions) continued to use DSs over time (i.e., persistent use). 

R2. If persistent Down syndrome is defined as reporting only two uses of Down syndrome, with an average of about 16 months, it should be compared with the definitions in other articles.

A: Chronic medical conditions are broadly defined by the Centers for Disease Control and Prevention (CDC) as “conditions that last 1 year or more and require ongoing medical attention or limit activities of daily living or both” (https://www.cdc.gov/chronic-disease/about/index.html).  Our study examined medical conditions that were diagnosed both 6 months before completion of the first questionnaire and between the two questionnaires, the latter an average of 1.3 years after the first questionnaire.  The study certainly falls within the CDC definition of a chronic condition in terms of temporality and requiring medical attention.   However, to the best of our knowledge, our study is the first study to examine persistent DS use in relation to CMCs so there are no other studies to which we can compare.  All previous studies have been cross-sectional.  We compare our results broadly to these cross-sectional studies in the Discussion.  

R3. The analysis is relatively simple, and the analysis of Down syndrome intake by age and sex for key chronic diseases can be analyzed.

A: We assume the reviewer is referring to Down syndrome as an example of a chronic medical condition.  In our study we examined a very broad array of medical conditions in relation to DS use.  Our definition of a chronic medical condition was one diagnosed in both phases of the study.  We controlled for the effects of age and sex (in addition to numerous other demographic and lifestyle characteristics) in our  multivariable (adjusted) analyses in Tables 3 and 4.

R4: References should be as new as possible.

A: We thoroughly reviewed the literature on DSs and medical conditions employing PubMed, Ovid, EBSCO, and Google Scholar by using relevant keywords to obtain appropriate information.  We are reasonably confident we have the most recently published journal articles, books, and websites on topics we were investigating.

Reviewer 2 Report

Comments and Suggestions for Authors

This study examines the relationship between the use of dietary supplements and the presence of chronic diseases in member of the US military. The study has a large sample size and appears to have been well carried out. The findings add to the literature on the use of supplements and reasons why people use them.

There are several issues that need attention.

p 2, line 52, the authors describe this study as a “longitudinal study”. While technically correct, this term will probably be misunderstood by most readers. Many readers are likely to fail to understand the difference between a longitudinal study and a prospective cohort study. It is clearer to describe this study as a repeated cross-sectional study. It is best to avoid using the word “longitudinal”.

The authors recorded DS use twice with a time gap of 1.3 years. The key feature of a cohort study is that at the time when the factor that may affect future health (eg sedentary behavior) is recorded none of the subjects have the outcome of interest (eg cancer). This requires a long time gap (years) between initial data collection and follow-up. With this methodology the findings can provide indications of a causal relationship. For example, sedentary behavior may predict an increased risk of cancer. This relationship may be causal or possibly that sedentary behavior is strongly associated with the real cause. As the sedentary behavior was recorded years before the cancer appeared, the temporal relationship is clear. In a cross-sectional study the sedentary behavior and the presence of cancer are recorded at the same time. The findings may indicate that sedentary behavior is associated with cancer but provide no indication of the temporal relationship. It may be that sedentary behavior leads to cancer or that cancer causes a person to cease engaging in exercise.

The study in this paper has little in common with a cohort study. A time gap of only 16 months is far too short for use of DS to prevent the CMCs being studied. Essentially they are recording use of DS and the presence of CMCs at roughly the same time. The findings do not allow inferences of a temporal relationship. For that reason the study is best described as a repeated cross-sectional study.

By far the most likely explanation for the findings is that people who have CMCs choose to use DS in an attempt to treat the condition. However, the data do not provide hard evidence to support this interpretation. It is also possible that in some cases the CMC was a side-effect caused by a harmful effect of a DS. For example, several herbal supplements can be toxic.

I strongly advice the authors to revise the paper based on the above. In particular, this must be stated on p 2, line 52, in the Discussion, and in Section 5 (Strength and Limitations).

A related issue is the use of data collected at the two time periods. It is not clear whether the findings are the average of the two sets of observations. As the proportion of subjects who provided data in the FU was only 25.3% of those in the BL (p 6, line 185), it is unclear how these findings were presented and interpreted.

Also in Section 5 (Strength and Limitations) there should be a comment on the response rate. This was only 18.2% (p 6, line 181). This is quite low and raises the real possibility of significant error due to self-selection bias. In particular, studies of this type are prone to the healthy volunteer effect (ie the people who volunteer tend to be healthier and have a greater interest in diet and health than the general population). The fact that there were more volunteers aged over 40 than under 30 (Table 2) also suggests a self-selection bias.

Author Response

(Reviewer’s comments preceded by an “R”; author’s replies preceded by an “A”)

R: This study examines the relationship between the use of dietary supplements and the presence of chronic diseases in member of the US military. The study has a large sample size and appears to have been well carried out. The findings add to the literature on the use of supplements and reasons why people use them.

A: Thank you for your summary and for taking time to review this work.  We address your comments and suggestions below.

R: p 2, line 52, the authors describe this study as a “longitudinal study”. While technically correct, this term will probably be misunderstood by most readers. Many readers are likely to fail to understand the difference between a longitudinal study and a prospective cohort study. It is clearer to describe this study as a repeated cross-sectional study. It is best to avoid using the word “longitudinal”.

The authors recorded DS use twice with a time gap of 1.3 years. The key feature of a cohort study is that at the time when the factor that may affect future health (eg sedentary behavior) is recorded none of the subjects have the outcome of interest (eg cancer). This requires a long time gap (years) between initial data collection and follow-up. With this methodology the findings can provide indications of a causal relationship. For example, sedentary behavior may predict an increased risk of cancer. This relationship may be causal or possibly that sedentary behavior is strongly associated with the real cause. As the sedentary behavior was recorded years before the cancer appeared, the temporal relationship is clear. In a cross-sectional study the sedentary behavior and the presence of cancer are recorded at the same time. The findings may indicate that sedentary behavior is associated with cancer but provide no indication of the temporal relationship. It may be that sedentary behavior leads to cancer or that cancer causes a person to cease engaging in exercise.

The study in this paper has little in common with a cohort study. A time gap of only 16 months is far too short for use of DS to prevent the CMCs being studied. Essentially they are recording use of DS and the presence of CMCs at roughly the same time. The findings do not allow inferences of a temporal relationship. For that reason the study is best described as a repeated cross-sectional study.

A: As the reviewer notes, this investigation is technically a longitudinal study because it involves two repeated observations of the same individuals over time on the same outcome(s).  In this case, the outcomes are dietary supplement use and clinically diagnosed medical conditions at two points in time.  The reviewer thinks we might more appropriately call this a repeated cross-sectional study.  However, repeated cross-sectional studies are those in which repeated observations are made on different samples of individuals on the same outcome(s) (see Caruana, J Thorac Dis 7(11):E537, 2015).  Thus, it is appropriate and correct to call this a longitudinal study.

As the reviewer also correctly notes, the definition of a cohort study is that participants do not have the outcome at baseline and are followed over time to get an estimate of how many people develop the outcome.  Our participants had both outcomes at baseline and follow-up and so calling the design a cohort study is not correct.  At no place in the manuscript have we referred to this investigation as a cohort study (retrospective or prospective).  We think this investigation is best referred to simply as a longitudinal study (i.e., same participants followed on two consecutive occasions) without reference to the term “cohort”.

R: By far the most likely explanation for the findings is that people who have CMCs choose to use DS in an attempt to treat the condition. However, the data do not provide hard evidence to support this interpretation. It is also possible that in some cases the CMC was a side-effect caused by a harmful effect of a DS. For example, several herbal supplements can be toxic.

A: Our data indicated that individuals with any CMC had 1.25 times higher odds of reporting DS use on both occasions after adjustment for demographic and lifestyle factors (Table 3).  Overall, persistent DS use was reported by 70% of individuals with CMC and 64% of individuals without CMCs.  Thus, participants with and without CMCs reported DS use on both occasions, but there was a higher reporting prevalence in those with CMCs.  As the reviewer notes, one hypothesis for the higher prevalence of persistent DS among those with CMCs is the use of supplements to treat their conditions and we have mentioned this possibility in the Introduction and extensively explored this possibility in the Discussion.  The reviewer notes that an alternate (or additional) explanation is that some DS may be causing CMCs and we have explored this possibility at length in other studies (Knapik, Food Chem Toxicol 161:112840, 2022; Knapik J Acad Nutr Dietet 122:1851, 2022).  Nonetheless, as the reviewer notes we cannot definitively determine with the data from this study if participants with CMCs have a higher prevalence of persistent DS use because they are using the DSs to treat/prevent conditions or if the DSs themselves are causing the conditions or both.  We have added to the Conclusion that the data supports the hypothesis that SMs may be using DS to self-treat medical conditions, although the possibility that the DSs may be causing eliciting the CMC may also be considered.

R: I strongly advice the authors to revise the paper based on the above. In particular, this must be stated on p 2, line 52, in the Discussion, and in Section 5 (Strength and Limitations).

A: For the reasons mentioned above we would like to continue to refer to this investigation as a “longitudinal study” without reference to the terms like “prospective”, “retrospective”, or “cohort”.  In the Discussion we have extensively considered the possibility that the DS may be used by some individuals to treat/prevent some specific CMCs and we have discussed the efficacy of DSs for treating/preventing specific CMCs. Most favorable effects are absent or relatively modest.  We have noted in the Strengths and Limitations section that persistent DS use and CMCs were recorded on only two occasions about 16 months apart and that a longer study (e.g., over several years) would have improved identification of longer-term DS users and CMCs of longer term.

R: A related issue is the use of data collected at the two time periods. It is not clear whether the findings are the average of the two sets of observations. As the proportion of subjects who provided data in the FU was only 25.3% of those in the BL (p 6, line 185), it is unclear how these findings were presented and interpreted.

A: The data are not an “average” of two sets of observations but rather discrete (individual) observations at the two periods.  Participants reported their DS use on a questionnaire on two occasions.  From a military medical surveillance system we collected all medical conditions occurring 6 months before the first questionnaire and in the periods between questionnaires.  We identified participants who did (and did not) report DSs on both questionnaires and individuals who had (and did not have) the general and specific medical conditions in both periods.    

R: Also in Section 5 (Strength and Limitations) there should be a comment on the response rate. This was only 18.2% (p 6, line 181). This is quite low and raises the real possibility of significant error due to self-selection bias. In particular, studies of this type are prone to the healthy volunteer effect (ie the people who volunteer tend to be healthier and have a greater interest in diet and health than the general population). The fact that there were more volunteers aged over 40 than under 30 (Table 2) also suggests a self-selection bias.

A: We have now included in the Strength and Limitations section that the response rate in the follow-up was only 25.3% of the baseline sample and we did not achieve the stratification we would have liked so there could be some bias (p 17, Lines 366-367). We have now also included in the first paragraph of the Results that compared to the requested stratified sample, the FU cohort was more likely to be female (12% vs 14%, p<0.01) and consisted of more Air Force personnel with fewer personnel from other services (Air Force 39%, Army 31%, Marine Corps 10%, Navy 19%, p<0.01).  While participation was not optimal, we do note that response bias was discussed in detail in a previous paper (p4, line 64-66). 

Reviewer 3 Report

Comments and Suggestions for Authors

This manuscript addresses an important topic related to the use of dietary supplements in association with prevention and/or treatment of medical conditions, particularly among individuals in the military. However, some changes need to be introduced before considering this manuscript:

. Title: What do authors mean by the word "persistent"? this should be clarified in the Methods section. 

. Abstract: Why are authors interested in exploring this association? suggest to add 1-2 background sentences to introduce the topic and the rationale of this study. Line 23, "SMs": authors should not start their sentences with abbreviations. When reporting the results, suggest that authors be more specific about the odds…and to add P values where needed. The conclusion is a repetition of the findings…it should focus more on the practical implications of these findings. 

. Keywords: Suggest to add keywords like dietary supplements, and US military.

. Introduction: Authors should also refer to studies stating that the US populations use DS to prevent diseases or manage diseases since their focus is on medical conditions. In the previous study conducted by authors, reference #28, what was the duration of DS use explored? and how is this study different from the previously conducted one? more information should be provided to support the rationale for this study. 

. Methods: In the Survey Description section, authors should clarify if and how this questionnaire was validated and/or tested for reliability before being implemented on a larger sample. 

. Results: In Table 2, authors should replace the word "gender" with "sex" as they are not exploring gender identity but rather physiological sex. Tables 3 and 4: authors should specify which ORs are significant…and add P values or highlight significant ones. Table 5: suggest to add to footnotes what a significant P value is. Overall, authors should clarify if there was any exploration of these findings by confounding variables such as sex; any differences in any of these findings for instance between men and women. 

. Discussion: Lines 261-268, citations need to be added. 

. References: extensive list and up to date. 

Comments on the Quality of English Language

Minor editing is needed in some parts of the manuscript. 

Author Response

See attached MSWord File

Round 2

Reviewer 2 Report

Comments and Suggestions for Authors

I stand by my comment that the term “longitudinal study” is often used to refer to cohort studies. Please do a search at Medline by entering that term as “title words”. You will then find quite a few cohort studies that are referred to as a “longitudinal study”. However, I don’t think that is a problem in your paper as you clearly describe your methodology at both the start of the Abstract and the start of the Methods section.

The authors appear to be suggesting that I view the study as being a cohort study and that the study should be labelled as such. That is not correct. I never made that suggestion.

The authors have largely ignored my final comment where I discuss the issue of response rate. It is well known that a low response rate may be a source of serious error in surveys. They mention the proportion of subjects who were female and the proportion who were in the air force. These aspects of possible limitations in the study are probably of minor importance. What is far more likely to be important is the low overall response at BL (18.2%) and the age distribution. I highlighted this in my previous comments. Unfortunately, the authors have failed to discuss this. Instead they say: “While participation was not optimal, we do note that response bias was discussed in detail in a previous paper.” It is very unlikely that readers of the paper will be familiar with a discussion of this issue in a previous paper. I therefore strongly encourage the authors to add a discussion of this issue.

Author Response

Reply to Second Round of Comments by Reviewer 1

(Reviewer’s comments preceded by an “R”; author’s replies preceded by an “A”)

R: I stand by my comment that the term “longitudinal study” is often used to refer to cohort studies. Please do a search at Medline by entering that term as “title words”. You will then find quite a few cohort studies that are referred to as a “longitudinal study”. However, I don’t think that is a problem in your paper as you clearly describe your methodology at both the start of the Abstract and the start of the Methods section.

A: My medical dictionary (Dorlands) defines a longitudinal study as “one in which participants, processes, or systems are studied over time, with data being collected at multiple intervals.”  We agree that longitudinal studies often refer to cohort investigations, but this is a subcategory of the larger rubric.  As the reviewer perceptively notes, we do make it clear in our Abstract and Methods how the data were obtained .  We thank the reviewer for understanding.

R: The authors appear to be suggesting that I view the study as being a cohort study and that the study should be labelled as such. That is not correct. I never made that suggestion.

A: We are very sorry for the misunderstanding.  We did not mean to say or imply that the reviewer saw this as a cohort study.  We were merely more fully defining a cohort study to note that WE did not see this as a cohort investigation.

R: The authors have largely ignored my final comment where I discuss the issue of response rate. It is well known that a low response rate may be a source of serious error in surveys. They mention the proportion of subjects who were female and the proportion who were in the air force. These aspects of possible limitations in the study are probably of minor importance. What is far more likely to be important is the low overall response at BL (18.2%) and the age distribution. I highlighted this in my previous comments. Unfortunately, the authors have failed to discuss this. Instead they say: “While participation was not optimal, we do note that response bias was discussed in detail in a previous paper.” It is very unlikely that readers of the paper will be familiar with a discussion of this issue in a previous paper. I therefore strongly encourage the authors to add a discussion of this issue.

            A: Response rates are not the only criteria on which to judge study quality and validity.  Details on how the sample differed from the requested sample frame is also important.  We agree with the reviewer that only a few of the most interested readers are likely to obtain the other papers to review the response bias. Thus, we have added the details regarding response bias from that paper to the Strengths and Limitations section on Lines 399-404.

            Rates of volunteerism for surveys in general have been falling in recent years and common explanations include “disillusionment with science and research, increased frequency of contacts by research groups, and increasing complexity of life in the 21st Century” (Morton, A NZ J Pub Health 36:106, 2012).  This is especially true in the military where Service Members are often requested to fill out surveys on various aspects of their service and suffer from a greater “survey burden” than their civilian counterparts (see https://apps.dtic.mil/sti/pdfs/AD1038400.pdf).  Between 2004 and 2018 response rates for Department of Defense active duty surveys have declined from 40% to about 15% (See https://www.opa.mil/research-analysis/methodology-studies/effect-of-declining-response-rates-on-opa-survey-estimates/).  Although by no means ideal, the 25% response rate we achieved in the present study (higher than most Department of Defense Surveys) may have been associated with the methods we used in an attempt to optimize participation.  As suggested by several reviews on web-based surveys (e.g., Sammut, Int J Nursing Studies 123:104058, 2021; Fan, Comp Hum Behav 26:132, 2010) response rates may be improved by 1) pre-notification of the impending survey, 2) a subject line saying the participants had been specifically selected for the study, 3) informative subject line in email requests, 4) salutation in invitations mentioning potential participant by name (first and last name), 5) use of only one personal ID number (PIN) to access survey (making access easy for participant while still identifying the participant), 6) reminders to complete survey by phone, email, or postal mail and 7) incentives such as cash or entry into a lottery if complete the survey (e.g., drawing for $1000).  Within the bounds of our time, budget, and what was allowed in the federal government and the military, we used the majority of these methods in an effort to maximize our response rate.  We could not phone the larger number of potential participants because of our budget and the intrusive nature of such contact, and we could not use the lottery incentive because of federal government restrictions on lotteries and gambling.